# Long-Term Outcomes in Severe Traumatic Brain Injury and Associated Factors: A Prospective Cohort Study

**DOI:** 10.3390/jcm11216466

**Published:** 2022-10-31

**Authors:** Daniel Vieira de Oliveira, Rita de Cássia Almeida Vieira, Leonardo Zumerkorn Pipek, Regina Marcia Cardoso de Sousa, Camila Pedroso Estevam de Souza, Eduesley Santana-Santos, Wellingson Silva Paiva

**Affiliations:** 1Hospital das Clínicas, Faculdade de Medicina FMUSP, Universidade de Sao Paulo, Rua Dr. Enéas de Carvalho Aguiar, 255, Sao Paulo 05403-010, SP, Brazil; 2School of Nursing, University of Sao Paulo, Sao Paulo 05403-000, SP, Brazil; 3Department of Statistical and Actuarial Sciences, University of Western Ontario, London, ON N6A 3K7, Canada; 4Nursing Department, University of Sergipe, São Cristóvão 49100-000, SE, Brazil

**Keywords:** severe traumatic brain injury, Glasgow Outcome Scale, recovery of function, outcome

## Abstract

Objective: The presence of focal lesion (FL) after a severe traumatic brain injury is an important factor in determining morbidity and mortality. Despite this relevance, few studies show the pattern of recovery of patients with severe traumatic brain injury (TBI) with FL within one year. The objective of this study was to identify the pattern of recovery, independence to perform activities of daily living (ADL), and factors associated with mortality and unfavorable outcome at six and twelve months after severe TBI with FL. Methodology: This is a prospective cohort, with data collected at admission, hospital discharge, three, six, and twelve months after TBI. RESULTS: The study included 131 adults with a mean age of 34.08 years. At twelve months, 39% of the participants died, 80% were functionally independent by the Glasgow Outcome Scale Extended, 79% by the Disability Rating Scale, 79% were independent for performing ADLs by the Katz Index, and 53.9% by the Lawton Scale. Report of alcohol intake, sedation time, length of stay in intensive care (ICU LOS), Glasgow Coma Scale, trauma severity indices, hyperglycemia, blood glucose, and infection were associated with death. At six and twelve months, tachypnea, age, ICU LOS, trauma severity indices, respiratory rate, multiple radiographic injuries, and cardiac rate were associated with dependence. Conclusions: Patients have satisfactory functional recovery up to twelve months after trauma, with an accentuated improvement in the first three months. Clinical and sociodemographic variables were associated with post-trauma outcomes. Almost all victims of severe TBI with focal lesions evolved to death or independence.

## 1. Introduction

Traumatic brain injury (TBI) occurs when a force transmitted directly to the brain results in pathophysiological damage and dysfunction that begins at the time of the accident and lasts for days to weeks [1]. TBI is the leading cause of death and disability in young people after trauma worldwide [2,3,4]. The global annual incidence has been estimated at 27.08 million, with an age-standardized incidence rate of 369 per 100,000 population [5]. A recent study shows a TBI incidence of 65.54 per 100,000 habitants in Brazil [6]. Although lesions after TBI are considered predominantly focal or diffuse in patients, most lesions identified in imaging exams are heterogeneous with the presence of both focal and diffuse components [7,8].

The consequences of the injury remain beyond the acute phase of TBI, extending and changing for a long time after the traumatic event [9,10]. The location and severity of the impact on the skull will determine the brain pathology and the neurological deficits caused by TBI [11]. Different clinical and sociodemographic variables have been related to unfavorable outcomes and functionality in patients with severe TBI, making it a relevant aspect in the research of the characteristics and pattern of recovery after the injury [12,13].

Thus, the recovery of patients with severe TBI is considered a complex and dynamic process, which requires early identification of consequences during the critical period of hospital stay and monitoring over time to prevent and manage clinical problems [14]. Referring those patients to specialized center is primordial [15]. Bonow et al. [3] evaluated the prognostic factors of 550 patients with TBI in Latin America. The authors observed that patients with subdural hematoma (SDH), intracerebral hematoma or contusion, and/or intraventricular hemorrhage had higher rates of hospital mortality and long-term disability, while those with epidural hematoma (EDH) had more favorable outcomes. Kvint et al. [14] found that early interval Computed Tomography (CT) imaging, clinical observation, the timing of surgical intervention, lesion location, intracranial pressure monitoring, surgical techniques, and reevaluation of surgical indication were associated with mortality and poor outcome in patients with intracerebral hemorrhage, SDH and EDH.

To assess other aspects of recovery after TBI, in addition to the Glasgow Outcome Scale Extended (GOS-E) and the Disability Rating Scale (DRS), which are already widely applied [16], some studies [17,18] analyzed the routine performance of these patients using the Activities of Daily Living (ADL) and Instrumental Activities of Daily Living (IADL) scale. Describing ADL and IADL is relevant to document the different levels of independence and deficits in carrying out daily activities according to severity, especially in the first few months after TBI [19,20,21,22,23]. Hammond et al. showed that more than half of the 110 individuals with disorders of consciousness (i.e., coma, vegetative state, minimally conscious state) due to moderate and severe TBI achieved ADL independence one year after injury, with progressive improvements over a 10-year follow-up [22].

Knowing the recovery pattern, ADL, and IADL of victims with severe TBI with focal lesions, and the factors associated with this outcome are important to identify the prognosis of these patients up to one year after TBI, provide resources for comprehensive care for these patients, evaluate the care provided, direct treatment to minimize the disabilities caused by TBI, guide the family as to the probable prognosis, in addition to contributing to the development of systematic assistance aimed at the rehabilitation and reintegration of victims into society [24].

Other studies also show a progressive improvement in recovery from hospital discharge up to ten years after TBI, however few of them have addressed long-term functional recovery and independence to perform ADL and IADL in patients with severe TBI with focal injury [19,22,25]. In this context, the objective of this study was to investigate factors in the initial patient care associated with the TBI prognosis in the short and long term.

## 2. Methods

### 2.1. Design

This is an observational prospective cohort study with patients with severe TBI and predominantly focal lesions such as contusion, subdural hematoma (SDH), epidural hematoma (EDH), and intraparenchymal hemorrhage.

### 2.2. Definitions

Severe TBI was considered as admission GCS ≤ 8. Traumatic subarachnoid hemorrhage (SAH) with concomitant focal lesion were also included. Those lesions can be the result of focal damage, but it can often be seen in more diffuse vascular lesions. Patients with exclusive SAH were excluded. There were five time points for data collection: during hospitalization, at hospital discharge, and three, six, and 12 months after the event.

### 2.3. Population

This study was carried out in the largest trauma hospital in Latin America. Hospital das Clínicas da Faculdade de Medicina da USP (HCFMUSP) has more than 50 surgical rooms and 1000 inpatients capacity, with a dedicated neurotrauma and critical care unit, located in the state of São Paulo, Brazil.

### 2.4. Inclusion and Exclusion Criteria

Subjects with GCS ≤8 at hospital admission and between 18 years and 60 years of age were selected for the study. Participants diagnosed with severe TBI with focal lesions were included in the research. The following criteria were used to exclude patients: initial hospital treatment after six hours of the traumatic event, transfer from other hospitals, previous diagnosis of TBI, exclusive diffuse lesions or with normal CT, presence of psychiatric disorders or chronic pathologies (e.g., acquired immunodeficiency, chronic kidney disease) and with lesions in the spinal cord region with severity ≥ 3, according to the AIS classification [26]. Patients with polytrauma were included in our study.

### 2.5. Ethical Aspects

The institutional ethics committee (Hospital das Clínicas da Faculdade de Medicina da USP, São Paulo, Brazil) approved the study and waived the need for patient informed consent. This study was carried out following the recommendations of the Declaration of Helsinki II [23].

### 2.6. Variables

Clinical variables related to trauma, hospitalization, and outcomes were recorded for each participant and prospectively collected, including mortality at 14 days, 6 months and 12 months.

#### 2.6.1. Exposures

The clinical variables related to trauma surveyed at hospital admission were: age, reported alcohol intake; Glasgow Coma Scale (GCS) [27]; Orotracheal intubation (OTI); respiratory rate (RR) (normal range of 12–20), cardiac rate (HR) (normal range of 50–100); hypotension (systolic pressure < 90 mmHg or diastolic < 60 mmHg); blood glucose (target range of 140 to 180 mg/dL); hypoxia (SpO2 < 90%) [28]; CT findings and number of brain lesions.

#### 2.6.2. Effect Modifiers

The variables identified during hospitalization related to hospitalization were: time sedated while mechanically ventilated, length of stay in the intensive care unit (LOS ICU), presence of any infection that required active treatment, presence of intracranial hypertension (ICH) assessed by clinical and radiological findings and neurosurgical treatment [29].

#### 2.6.3. Outcomes

Trauma indices assess the global severity of injuries in individuals after a traumatic event. The Injury Severity Score (ISS) scales were used for the main article. Data for the following scales are available in the Appendix A: New Injury Severity Score (NISS); Revised Trauma Score (RTS); Trauma and Injury Severity Score (TRISS); New Trauma and Injury Severity Score (NTRISS) [30,31,32,33,34,35].

The ISS is an anatomical trauma index that assesess the severity of injuries from the sum of squares of the 3 highest Abbreviated Injury scores Scale (AIS). ISS scores range from 1 to 75, the higher the score, the greater the severity of the injuries.

Regarding outcome-related variables, mortality and functional dependence at six and 12 months with GOS-E were used. Due to the severity of TBI and the multidimensional nature of recovery, it was decided to use GOS-E to assess functional dependence. The DRS scale is also available as Appendix A [36].

The GOS-E score ranges from one to eight, being: (1) dead, (2) vegetative state, (3) lower severe disability, (4) upper severe disability, (5) lower moderate disability, (6) upper moderate disability, (7) lower good recovery, (8) upper good recovery. Scores from 1 to 4 were classified as dependent and scores from 5 to 8 as independent [37].

Independence for ADL was assessed using the Katz index and IADL was assessed using the Lawton scale. The Katz Index assesses six ADL functions: bathing; feeding; dressing up; going to the bathroom; getting in and out of bed and maintaining sphincter control [35]. Katz’s ADL scores Index ranges from zero to 18 and, the lower the score, the greater the independence for ADL [38]. The Lawton Scale classifies the individual’s ability to perform IADL through the evaluation of eight domains, these scored in one (unable), two (needs assistance) and three (able). The IADL includes actions such as preparing your own food, shopping, moving around, cleaning the house, and managing your own money [39]. The sum of the scores can range from eight to 24, with higher scores for the ability to carry out IADL independently [40].

### 2.7. Statistical Analysis

Data is presented as mean and standard deviation (SD) for continuous variables, median with IQRs for ordinal variables and count for categorical variables. Shapiro–Wilk test was used to examine normality of distribution of continuous variables. To compare the average of the total value for GOS-E, DRS, Katz Index, and Lawton Scale between time points, the non-parametric Quade test was used. As there was a significant difference between the results of these evaluations, multiple comparisons were made (two-by-two comparisons between times), using the Wilcoxon signed-rank test with correction for multiple comparisons with the Holm–Bonferroni method.

To identify associations between variables of interest and death outcomes up to twelve months and recovery aspects (functional dependence) at six and twelve months after trauma, comparisons were made between groups of people who died or not, and between those who were dependent or independent at six and twelve months after severe TBI with focal lesions. In these comparisons, Pearson’s Chi-Square or Fisher’s Exact test, Student’s *t*-test for two samples, Wilcoxon-Mann–Whitney test, and Brunner-Munzel test were applied.

All information collected was stored in a computerized database built using the R 3.4.2 software (packages tidyverse, ggpubr, rstatix and readxl). For all analyses, a significance level of 5% was considered. The analysis was blindly performed by the author C.P.E.d.S.

## 3. Results

### 3.1. Patient Characteristics

Between September 2014 and September 2017, 319 patients with GCS ≤ 8 were admitted. Of these, 133 were included based on the inclusion and exclusion criteria. Figure 1 shows the flowchart of patient selection. Detailed demographic information is presented in Table 1.

The most frequent injuries were cerebral contusion (29%), subdural hematoma (28%), subarachnoid hemorrhage (20%), and extradural hematoma (16%).

Regarding the severity of the trauma, the ISS ranged from 14 to 66, with a mean of 33.17 (SD = 8.785) and a median of 35; the ISS ranged from 14 to 75, with a mean of 53.27 (SD = 15.212) and a median of 57.

The mean sedation time was 6.4 days (SD = 7.16). Almost all victims (97%) in this study were admitted to the intensive care unit (ICU). The length of stay in the ICU ranged from one to 71 days. The mean length of stay in the ICU was 13.8 days (SD = 13.42), with a median of 10.5 days. During the hospital stay, 47 patients (35.3%) developed an infection and 18 patients (13%) developed intracranial hypertension.

During hospital stay, 63 victims (47%) underwent neurosurgical procedure. Of these, the most frequent procedure was decompressive craniotomy, performed in 84.1% of the patients. Among surgical procedures performed by other specialties, 11 patients (8%) had orthopedic procedures and ten patients (7%) had general surgery procedures. The hospital length of stay varied between one and 187 days, with a mean of 24.78 days (SD = 29.43) and a median of 14 days.

### 3.2. Outcome

Table 2 shows a progressive improvement in terms of functional recovery in all scales used, being more expressive in the first three months after the trauma. A total of 51 patients (38%) died within 12 months of follow-up (40 died within the first 14 days). Of the 82 surviving patients, 75 patients (91%) participated in the entire follow-up and 7 (8%) participated only in part of the follow-up (discharge).

At discharge, 50 (61%) patients were classified as having a severe disability and no participant was in a vegetative state. After 12 months, 15 participants were dependent, nine (12%) had severe disability, and 6 participants (8%) had a vegetative state (Figure 2).

The results of multiple comparisons indicate that the categorization of GOS-E was statistically different between the periods consisting of discharge and three months, the period between three months and six months, the period of discharge and twelve months, discharge and six months (*p* < 0.05). There was no statistically significant difference in the period between 6 and 12 months (*p* = 0.061) after trauma.

For ADLs described in Table 2, it was found that the mean score changed from 9.22 (SD = 7.28) at discharge to 3.09 (SD = 6.39) after 12 months. There was a statistically significant difference between the Katz index scores at discharge, three, six, and twelve months after trauma. It was found that the mean score, standard deviation, and median score showed a steeper drop between discharge and assessment at three months. This finding is similar to the findings of the GOS-E and DRS scores. In two-by-two multiple comparisons, a statistically significant difference was observed between the averages of the Katz index values in all compared periods (*p* < 0.05).

Table 3 shows the average score of the six domains evaluated by the Katz Index in the four study periods. There was a decrease in the average score in all items of the scale for all periods. It is also noted that the items ‘bathing’ and ‘dressing’ initially showed a more accentuated decline. The domain ‘feeding’ had the most constant improvement rate, with the highest score at discharge and the best performance at 12 months.

Regarding IADL, Table 2 shows the average score using the Lawton scale, ranging from 17.01 (SD = 6.65) at 3 months to 19.91 (SD = 5.98) at 12 months. Table 3 shows the average score in the eight domains evaluated by the Lawton scale in the three periods of outpatient observation. It is observed that the increase in the average score occurred in almost all items of the scale, except for the “ability to handle finances”, which deteriorated relative to the average scores. The domain with the best improvement was ‘Laundry’, with an average increase of 0.50, followed by ‘shopping’ and ‘Food preparation’, with an average increase of 0.38 each.

### 3.3. Predictors of Mortality and Disability

Within 14 days after TBI with focal lesions, 40 participants (30%) had died, and 82 patients were discharged. The main cause of death was brain death in 55% of patients, cardiopulmonary arrest in 22.50%. After six months, 19 participants (25%) were classified as dependents according to GOS-E (3 participants with severe upper disability, 10 participants with severe lower disability and 6 participants in a vegetative state). At 12 months, 15 participants (20%) were classified as dependent (5 participants with upper severe disability, 4 participants with lower severe disability, and 6 participants in vegetative state).

Table 4 and Table 5 show the factors associated with death within 14 days and dependence at 6 and 12 months. There is an important difference between patients who died or not within 14 days in relation to sedation time (*p* < 0.001), length of stay in ICU (*p* < 0.001), GCS (*p* < 0.001), reports of alcohol consumption (*p* = 0.029), glycemic alterations (*p* < 0.01), hyperglycemia (*p* = 0.039) and the occurrence of infection (*p* < 0.01). ISS, NISS, and RTS and survival prediction models (TRISS and NTRISS) also showed a statistically significant association with 14-day death (*p* < 0.005).

Age (*p* = 0.003), length of stay in the ICU (*p* < 0.001), changes in respiratory rate (*p* = 0.048), and two or more brain radiographic injuries (*p* = 0.011) were factors associated with dependence at six months. Similarly, age (*p* = 0.009), length of stay in the ICU (*p* < 0.001), changes in heart rate (*p* = 0.028), two or more brain radiographic injuries (*p* = 0.001) were factors associated with dependence at 12 months. As for the trauma severity indices, only the NISS (*p* = 0.026) was associated with dependence at six months, and the NISS (*p* = 0.002), TRISS (*p* = 0.019), and NTRISS (*p* = 0.016) at twelve months.

Due to the different nature of lesion from the patients in our study, Table 6 shows the outcome in 14 days, 6 months, and 12 months for each TBI lesion (epidural hematoma, focal lesion without epidural hematoma, subdural hematoma and contusion). There was no statistically significant difference between the groups (*p* > 0.05).

## 4. Discussion

Considering that TBI is the main cause of disability and absence from productive activity [9,41], there is a continuous concern to know the recovery curves that represent improvements after severe TBI and their factors. Identifying the evolution, functionality, and independence to perform ADL allows the medical team and the patient to know the recovery pattern and even the stability of the brain injury. In our investigation, high mortality in patients was evidenced, however, most survivors showed a progressive improvement in functional recovery and independence from ADL up to 12 months, with a more pronounced improvement in the first three months.

We observed in our cohort after severe TBI with focal lesions, 40 participants (30.1%) died within 14 days and 51 participants (38%) within 12 months. Due to the severity of TBI, most participants die within the first few days after the trauma. The main cause of death was brain death in 55% of patients and cardiopulmonary arrest in 22.50%. All those patients were treated in the largest and one of the most important trauma hospital in Latin America and died despite the best possible treatment, including a specialized trauma and intensive care team, early imaging (less than 30 min after arriving at the hospital), and immediate surgery, if necessary. Moreover, the high mortality in those patients is also related to the initial very severe injury (39 out 40 patients had an ISS score > 25, *p* = 0.006). Research carried out in Latin America with severe TBI showed similar results, with 28% of participants dying during hospitalization, while 36% died in the first six months of the study [3].

Regarding the recovery pattern in our study, there was a progressive improvement in functionality in the first 12 months after severe TBI. Previous prospective observations indicate that, in the first six months after TBI, great changes occur in functional recovery and last until the period of five to ten years after the trauma [11,42]. It was also found that, among the survivors evaluated up to twelve months (75 participants), 80% achieved categorization in the GOS-E consistent with independent living and 40% of them showed complete TBI recovery. Our results were similar to previous data [43], however other studies [44,45] show levels of dependence at twelve months higher than those evidenced in our research. Unlike our findings, a prospective study conducted with participants with severe TBI showed a higher frequency of dependence twelve months after trauma, where 40% were classified as dependent, while 18% of participants had an upper good recovery or lower good recovery by GOS-E.

Improvement in independence was also observed by DRS, especially in the first three months after the trauma. DRS is used to measure disability during rehabilitation and is considered by some authors to be less sensitive than GOS-E to assess functionality. On the other hand, this tool allows observing small functional changes in the acute phase after trauma that are not detected by the dichotomous version of the GOS. In addition to being another measure to assess functionality in participants with severe TBI, the DRS has eight ranges of values for vegetative state, fifteen ranges of values between moderately severe and extremely severe disability, and six ranges of values for mild to moderate disability [46]. The DRS scores showed a mean of 18.47 (SD = 10.98) at discharge, classified as extremely severe disability, changing to a mean score of 7.97 (SD = 8.33) at three months (moderately severe). At six months, the mean score was 5.49 (SD = 7.83), which would already classify the participants as independent, and, at twelve months, a mean score of 4.21 (SD = 6.9) [46,47,48].

In relation to ADL, there was a decrease in the average Katz scale score in all periods. These differences in the mean scores indicated that there was an improvement in ADL between the four intervals of assessment of the victims, being more pronounced between the observations at three and six months. There was a tendency to stabilize at 12 months. A similar was observed using the Lawton scale. After twelve months, most victims were classified as independent by GOS-E (80%), DRS (79%), and Katz Index (79%) assessments, while half of the participants were classified as independent by the Lawton Scale (54%).

Studies with severe TBI victims have not used repeated measures of ADL and IADL capacity in the first months after trauma to characterize the course of spontaneous remission of the consequences of this injury. For this purpose, the use of functionality scales that seek to cover the global outcome of disabilities, such as GOS, GOS-E, DRS, and the Functional Independence Measure (FIM), allows the assessment of the individual as a whole and integrated into society [11,19,25,36,37,49,50,51,52,53]. Indeed, IADL assesses more complex domains than ADL, which involves more than only physical skills that enable daily activities. IADL evaluates cognitive abilities, such as memory, attention, planning, functions to do shopping, food preparation, responsibility for your own medications, and ability to handle finances. Therefore, the results observed in the literature show that cognitive alterations after severe TBI can be predictors of individual functional independence, productivity and return to society [54,55,56]. In particular, the knowledge of these alterations helps the multidisciplinary team to direct the appropriate treatment at specific moments, in order to obtain a more effective rehabilitation.

In our study, approximately half (54%) of the participants were independent to perform IADL at twelve months after severe TBI. Although the number of patients who underwent treatment with other specialists (physiotherapy, speech therapy, occupational therapy, psychologist) varied between 56% at three months to 47% at twelve months, the severity of the injury caused by TBI does not reach the final pattern of recovery and independence to perform more complex activities of daily living in the first twelve months in part of the individuals. This improvement in recovery can persist in some individuals, with varying degrees of intensity, for two to ten years after trauma [57]. A study with 110 individuals with traumatic disorders of consciousness due to moderate and severe TBI showed that more than half of the individuals achieved near-maximal recovery by one year and that the proportion of participants achieving functional independence increased between 5 and 10 years post-injury [22]. Two recent studies with a population of patient with moderate to severe TBI shows a similar pattern of recover in the long term, highlighting the importance of not making early, definitive prognostic statements [58,59].

Previous research has shown that sociodemographic and physiological variables are associated with mortality and unfavorable outcome after severe TBI [3,44,60,61,62,63,64]. Thus, trauma severity indices represent one of the most important risk factors for mortality and dependence after severe TBI. Most victims of severe TBI with focal lesions had more than one brain injury (67%) and more than a quarter of these victims (26%) had three or more intracranial lesions. Among them, cerebral contusion was the most frequent finding (29%), followed by subdural hematoma (28%).

Surprisingly, two or more brain radiographic injuries were not associated with death within 14 days; however, there was an association between these findings and dependence at six and twelve months. Hilario et al. [65] observed that individuals with hemorrhagic brain radiographic injuries, bilateral injuries, and posterior lesions are not related to worse prognoses at six months after severe TBI, whereas other studies show that individuals with head AIS severity > 5 with large (>1 cm thick), intracranial pathology diagnosed by CT scan, underwent ICP monitoring, non-evacuated mass lesions, lower GCS motor, and cisternal effacement had poor six-month GOS-E [3,66]. In a recent meta-analysis, Zhang et al. [67] showed that decompressive craniotomy in individuals with TBI could significantly reduce mortality rate, lower intracranial pressure (ICP), decrease the LOS ICU, and hospital stay, but also increase complications rate. Previous studies have shown similar results that the severity of the injury of patients is associated with early mortality, poor outcome, and dependence in patients with TBI [61,62,63,64,68,69]. Regarding TBI lesion, patients with SDH had a tendency for worse outcomes when compared to other lesions, as previous described in the literature [70]. Yuh et al. also use data from the studies TRACK-TBI and CENTER-TBI and showed that specific pathological CT features carried different prognostic implications after mTBI to 1 year postinjury [71]. Despite the different physiopathology and natural history of disease, this difference was not statistical significant when comparing any type of TBI lesion in our study (EDH, SDH, focal lesion without EDH and contusion).

Secondary injury is a complex process that occurs in the hours and days after primary injury, which encompasses cranial and systemic complications. Matovu et al. [72] showed that 1 in 6 patients with severe TBI admitted to a hospital in Uganda had blood glucose levels higher than 11.1 mmol/L. Patients with hyperglycemia were 1.47 times more likely to die within 30 days compared to those with no hyperglycemia, but this association was not statistically significant (OR: 1.47; 95% CI [0.236–9.153], *p* = 0.680). Tohme et al. [68] showed that GCS < 9 in emergency department, abnormal pupil reaction, and ISS ≥ 25 were factors associated with mortality at 14 days after severe TBI (Abbreviated Injury Scale score of head region > 3). Likewise, in the Leskovan study et al. [73], age, ICU, GCS, ISS, LOS (day) are factors associated with mortality in patients those patients, however, the use of blood alcohol content was not a significant predictor of mortality at discharge. Finally, Bonow et al. was also able to show in his study in Latin America that the location for the patient treatment was also related with their outcome [3].

Despite data from 33.1% of patients being available only after prehospital intubation, respiratory rate was an independent predictor of poor outcome at 6 months in our cohort. Regarding oxygen levels, Volpi et al. [74] showed that hypoxia was associated with a poor outcome at six months, while Wahlin et al. [69] observed that prehospital hypoxia was not significantly correlated with prehospital intubation in univariate analysis for unconscious patients, and prehospital reports revealed that hypoxia was not the primary cause of prehospital intubation. It is important to note that despite the long prehospital transport time caused by the distance from the event to the hospital and the chaotic traffic in the city of São Paulo, 40% of the participants in our research were treated via air medical prehospital and with a recommendation of endotracheal intubation procedure to be performed when the individual has a GCS ≤ 8 (unconscious), as is suggested by the Prehospital trauma life support [69].

The present study has several limitations. The major limitation of this study is the absence of records on the clinical conditions in the scene, which limits the identification of factors associated with the TBI outcome. Secondly, the sample of this research included participants from a single institution, a reference center for the care of highly complex cases, bringing restrictions to the generalization of the results. Data for specific scales that evaluate spinal cord lesions were not collected. The correlation found between mortality and sedation may provide limited information because patients with early death cannot have prolonged sedation. Finally, the limited access to rehabilitation resources after discharge (physiotherapy, occupational therapy, and psychology) by nearly half of the survey participants can compromise functional improvement and independence for ADL over time. Moreover, specific data for which patient were able to have physical therapy and their outcome would also be an important information and necessary variable to precisely predict long term rehabilitation.

## 5. Conclusions

Mortality in patients with severe TBI with focal lesion in our trauma reference center in Latin America was high; however, most survivors reached conditions compatible with independent living at twelve months. During this period, the recovery of victims was accentuated in the first three months. Almost all victims of severe TBI with focal lesions evolved to death or independence. There was an improvement in functional capacity by GOE-E in 80% of those evaluated and in ADL performance in 93% of cases. Reported alcohol intake, sedation time, ICU LOS, GCS, trauma indices, hyperglycemia, blood glucose, and infection were associated with death. Tachypnea (RR > 20), age, ICU LOS, NISS, TRISS, NTRISS, two or more brain radiographic injuries, and cardiac rate were associated with dependence at six months and twelve months. This data can help health providers better predict outcomes and provide adequate health treatment.

## Figures and Tables

**Figure 1 jcm-11-06466-f001:**
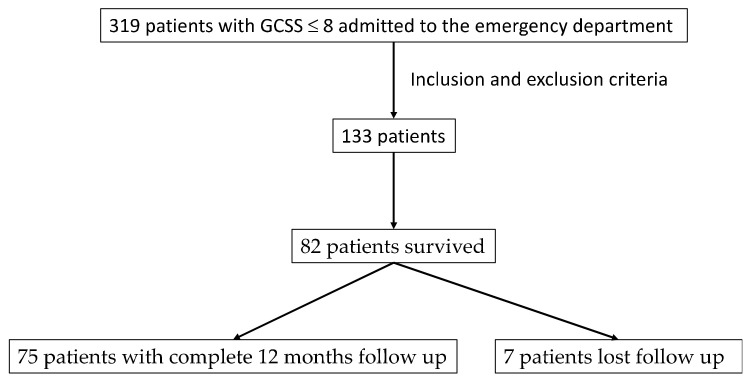
Flowchart on sample selection.

**Figure 2 jcm-11-06466-f002:**
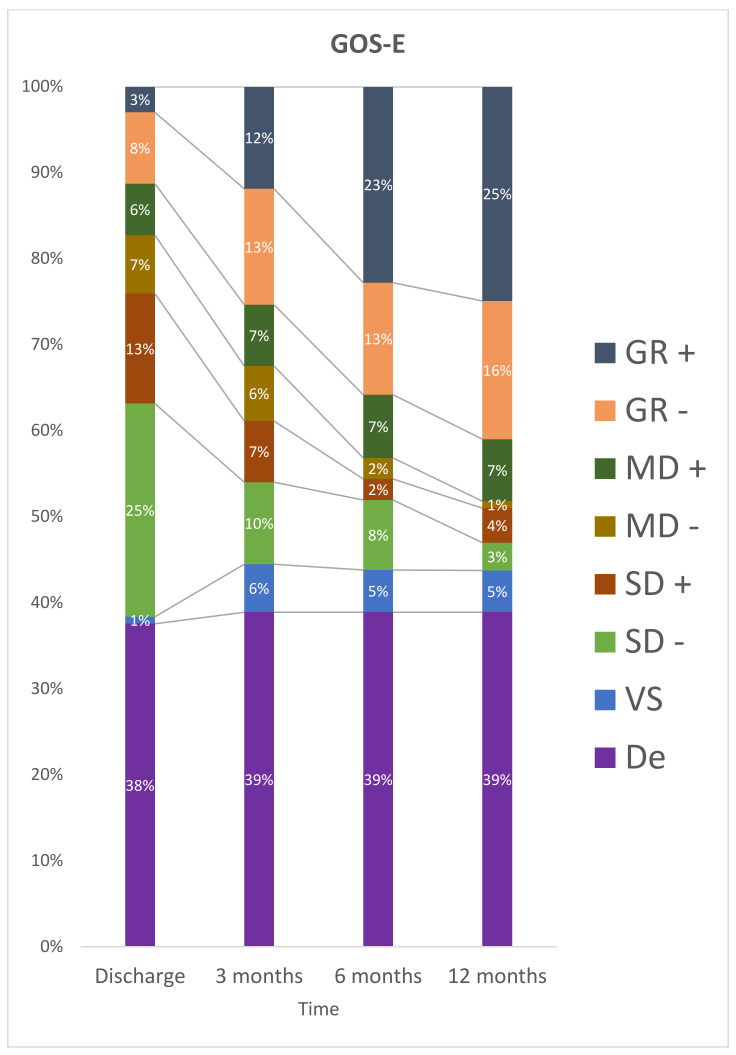
Stratification for GOS-E at discharge, 3 months, 6 months and 12 months after TBI. De: Death; GR– : Lower Good Recovery; GR+: Upper Good Recovery; MD–: Lower Moderate Disability; MD+: Upper Moderate Disability; SD–: Lower Severe Disability; SD+: Upper Severe Disability; VS: Vegetative State.

**Table 1 jcm-11-06466-t001:** Demographic data for patients at admission.

Variable			*n*	%
Age	mean		34.08 (SD 12.35)	
Sex	male		117	88%
	female		16	12%
Race	white		86	65%
	mixed		29	22%
	other		17	13%
Etiology	transport accidents	pedestrians	25	19%
		motorcycle	22	16%
		automobile	12	9%
		cycling	4	3%
		other	28	21%
	falls		33	25%
	aggression		5	4%
	other trauma		5	4%
Alcohol consumption	Yes		12	19%
	No		53	81%
Number of brain radiographic injuries	1		44	33%
	2		55	41%
	3		28	21%
	≥4		6	5%
Respiratory frequency	<12		1	1%
	12–20		101	77%
	>20		30	23%
Hypoxia	Yes		13	11%
	No		110	89%
Orotracheal intubation	Yes		44	33%
	No		89	67%
Sedation	Yes		118	89%
	No		15	11%
Heart frequency	<50		3	2%
	50–100		80	62%
	>100		46	36%
Hypotension	Yes		10	8%
	No		123	92%
Pupillary changes	Yes		77	58%
	No		56	42%

Data presented and number of patient and percentage.

**Table 2 jcm-11-06466-t002:** Comparison of GOS-E, Katz and Lawton scale Index scores after trauma in severe TBI victims with focal injuries.

Scale		Mean (SD)	Median	Min–Max	*p*-Value
GOS-E	Discharge	2.52 (1.642)	2	1–6	<0.001
	3 months	4.51 (2.008)	5	1–7	
	6 months	5.16 (2.075)	6	1–7	
	12 months	5.49 (1.894)	6	1–7	
Katz	Discharge	9.22 (7.28)	10	0–18	<0.001
	3 months	5.66 (7.1)	1	0–18	
	6 months	4.15 (6.95)	0	0–18	
	12 months	3.09 (6.39)	0	0–18	
Lawton	3 months	17.01 (6.65)	19	8–24	<0.001
	6 months	19.25 (6.45)	24	8–24	
	12 months	19.91 (5.98)	24	8–24	

SD—Standard deviation; Max—maximum value; Min—minimum value; GOS-E—Glasgow Outcome Scale Extended; Participants during follow-up: 82 participants at discharge, 77 participants at three months, 75 participants at six months and 76 participants at twelve months.

**Table 3 jcm-11-06466-t003:** Comparison of ADL and IADL domains between hospital discharge and twelve months for victims of severe TBI with focal lesions.

Scale	Domain	Time	Mean	SD	Median	Min–Max	*p*-Value
ADL KATZ INDEX	Bathing	Discharge	1.78	1.18	2	0–3	<0.001
		3 months	1.09	1.25	0	0–3	
		6 months	0.77	1.19	0	0–3	
		12 months	0.55	1.12	0	0–3	
	Dressing	Discharge	1.69	1.24	2	0–3	<0.001
		3 months	1.06	1.23	0	0–3	
		6 months	0.75	1.21	0	0–3	
		12 months	0.55	1.12	0	0–3	
	Toileting	Discharge	1.66	1.26	2	0–3	<0.001
		3 months	0.94	1.27	0	0–3	
		6 months	0.63	1.19	0	0–3	
		12 months	0.49	1.10	0	0–3	
	Transferring	Discharge	1.60	1.27	2	0–3	<0.001
		3 months	0.94	1.21	0	0–3	
		6 months	0.79	1.20	0	0–3	
		12 months	0.54	1.10	0	0–3	
	Continence	Discharge	1.47	1.41	1	0–3	<0.001
		3 months	0.81	1.25	0	0–3	
		6 months	0.63	1.17	0	0–3	
		12 months	0.53	1.14	0	0–3	
	Feeding	Discharge	1.24	1.32	1	0–3	<0.001
		3 months	0.83	1.29	0	0–3	
		6 months	0.59	1.16	0	0–3	
		12 months	0.43	1.06	0	0–3	
IADL LAWTON SCALE	Use Telephone	3 months	2.38	0.87	3	1–3	<0.001
		6 months	2.61	0.77	3	1–3	
		12 months	2.64	0.71	3	1–3	
	Mode of Transportation	3 months	2.25	0.85	3	1–3	<0.001
		6 months	2.49	0.79	3	1–3	
		12 months	2.54	0.76	3	1–3	
	Shopping	3 months	2.05	0.94	2	1–3	<0.001
		6 months	2.37	0.88	3	1–3	
		12 months	2.43	0.82	3	1–3	
	Food Preparation	3 months	2.04	0.97	2	1–3	<0.001
		6 months	2.32	0.89	3	1–3	
		12 months	2.42	0.82	3	1–3	
	Housekeeping	3 months	1.92	0.94	2	1–3	<0.001
		6 months	2.28	0.91	3	1–3	
		12 months	2.42	0.85	3	1–3	
	Laundry	3 months	1.95	0.94	2	1–3	<0.001
		6 months	2.31	0.91	3	1–3	
		12 months	2.45	0.85	3	1–3	
	Responsibility for Own Medications	3 months	2.28	0.91	3	1–3	<0.001
		6 months	2.41	0.81	3	1–3	
		12 months	2.49	0.81	3	1–3	
	Ability to Handle Finances	3 months	2.35	0.89	3	1–3	<0.001
		6 months	2.56	0.84	3	1–3	
		12 months	2.51	0.81	3	1–3	

ADL—Activities of daily living; IADL—Instrumental activities of daily living; SD—Standard deviation; Max—maximum value; Min—minimum value. Comparison using Quade test. Post hoc with Wilcoxon and correction with Holm.

**Table 4 jcm-11-06466-t004:** Factors associated with 14-day mortality in severe TBI victims with focal lesions.

Associated Factors		14-Day Mortality		*p*-Value
		Yes	No	
Age	Mean (SD)	35.04 (13.19)	33.14 (11.57)	0.47 *
	Min–Max	18–60	18–60	
Sedation time	Mean (SD)	1.75 (2.38)	6.61 (5.97)	<0.001 ****
	Min–Max	0–8	0–34	
ICU LOS	Mean (SD)	9.92 (14.23)	24.81 (75.59)	<0.001 ****
	Min–Max	0–71	0–675	
GCS	Mean (SD)	3.63 (1.28)	4.72 (2.0)	<0.001 *
	Min–Max	3–8	3–8	
ISS	Mean (SD)	35.43 (8.38)	31.62 (8.75)	0.032 *
	Min–Max	17–66	14–50	
Gender	Male	47 (39.83)	71 (60.17)	0.415 **
	Female	4 (28.57)	10 (71.43)	
Alcohol intake (related)	Yes	4 (15.38)	22 (84.62)	0.029 **
	No	16 (41.03)	23 (58.97)	
Intubation	Yes	17 (39.53)	26 (60.47)	0.883 **
	No	34 (38.2)	55 (61.8)	
Sedation	Yes	47 (39.83)	71 (60.17)	0.526 **
	No	4 (30.77)	9 (69.23)	
Pupillary changes	Yes	33 (42.86)	44 (57.14)	0.240 **
	No	18 (32.73)	37 (67.27)	
Respiratory rate	<12 RR	0 (0)	1 (100)	0.898 ***
	12–20 RR	38 (38)	62 (62)	
	>20 RR	12 (40)	18 (60)	
Cardiac Rate	<50 BPM	1 (33.3)	2 (66.67)	0.916 **
	50–100 BPM	30 (37.97)	49 (62.03)	
	>100 BPM	19 (41.3)	27 (58.7)	
Hypotension	Yes	4 (40)	6 (60)	0.927 **
	No	47 (38.52)	75 (61.48)	
Hypoxia	Yes	7 (53.85)	6 (46.15)	0.354 **
	No	44 (40.37)	65 (59.63)	
Blood glucose	70–200 mg/dL	27 (39.13)	42 (60.87)	<0.01 ***
	>200 mg/dL	16 (84.21)	3 (15.79)	
	<70 mg/dL	1 (50)	1 (50)	
Hypoglycemia	Yes	1 (16.67)	5 (83.33)	0.336 **
	No	41 (35.96)	73 (64.04)	
Hyperglycemia	Yes	29 (42.65)	39 (57.35)	0.039 **
	No	13 (24.53)	40 (75.47)	
number of brain radiographic injuries	≥2	26 (33.33)	52 (66.67)	0.134 **
	1	25 (46.3)	29 (53.7)	
Neurosurgical procedure	Yes	26 (41.94)	36 (58.06)	0.465 **
	No	25 (35.71)	45 (64.29)	
Infection	Yes	7 (15.22)	39 (84.78)	<0.01 **
	No	38 (48.72)	40 (51.28)	
Intracranial Hypertension	Yes	6 (33.33)	12 (66.67)	0.592 **
	No	44 (40)	66 (60)	

BPM—beats per minute; GCS—Glasgow Coma Scale; ICU—Intensive Care Unit; ISS–Injury Severity Score; LOS—Length of stay; Max—Maximum value; mg/dL—milligrams per deciliter; Min—Minimum value; RR—respiratory rate; SD—Standard deviation. * Wilcoxon-Mann–Whitney; ** Pearson’s chi-squared test; *** Fisher test; **** Brunner-Munze test.

**Table 5 jcm-11-06466-t005:** Factors associated with functional dependence at 6 and 12 months in severe TBI victims with focal lesions.

Associated Factors		6 Months Disability		*p*-Value	12 Months Disability		*p*-Value
		Yes	No		Yes	No	
Age	Mean (SD)	39.37 (11.46)	30.77 (10.82)	0.003	40.07 (12.04)	31.39 (10.84)	0.009 *
	Min–Max	19–60	18–60		19–60	18–60	
Sedation time (days)	Mean (SD)	8.16 (10.41)	6.91 (6.33)	0.995	9.4 (11.41)	6.58 (6.18)	0.458 ****
	Min–Max	0–46	0–34		0–46	0–34	
ICU LOS	Mean (SD)	29.44 (14.91)	25.35 (90.44)	<0.001	32.4 (14.48)	24.4 (87.3)	<0.001 ****
	Min–Max	0–46	0–34		17–66	0–675	
GCS	Mean (SD)	4.32 (1.63)	4.75 (2.03)	0.515	4.4 (1.72)	4.67 (1.99)	0.684 *
	Min–Max	3–8	3–8		3–8	3–8	
ISS	Mean (SD)	34.11 (7.48)	31.54 (9.05)	0.269	35.6 (6.34)	31.18 (9.05)	0.079 *
	Min–Max	17–45	14–50		21–45	14–50	
Gender	Male	16 (24.62)	49 (75.38)	0.717	12 (18.18)	54 (81.82)	0.385 **
	Female	3 (30)	7 (70)		3 (30)	7 (70)	
Alcohol intake (related)	Yes	7 (35)	13 (65)	0.255	5 (25)	15 (75)	0.595 **
	No	4 (19.05)	17 (80.95)		4 (18.18)	18 (81.82)	
Intubation	Yes	8 (33.33)	16 (66.67)	0.278	7 (29.17)	17 (70.83)	0.163 **
	No	11 (21.57)	40 (78.43)		8 (15.38)	44 (84.62)	
Sedation	Yes	19 (28.36)	48 (71.64)	0.180	15 (22.06)	53 (77.94)	0.333 **
	No	0 (0)	7 (100)		0 (0)	7 (100)	
Pupillary changes	Yes	12 (28.57)	30 (71.43)	0.470	11 (26.19)	31 (73.81)	0.119 **
	No	7 (21.21)	26 (78.79)		4 (11.76)	30 (88.24)	
Respiratory rate (RR)	<12	0 (0)	1 (100)	0.048	0 (-)	0 (-)	0.063 ***
	12–20	38 (38)	62 (62)		9 (14.75)	52 (85.25)	
	>20	12 (40)	18 (60)		6 (40)	9 (60)	
Cardiac Rate (bpm)	<50	0 (-)	0 (-)	0.059	2 (100)	0 (0)	0.028 **
	50–100	12 (20)	48 (80)		9 (19.57)	37 (80.43)	
	>100	7 (46.67)	8 (53.33)		3 (12)	22 (88)	
Hypotension	Yes	1 (16.67)	5 (83.33)	0.613	1 (16.67)	5 (83.33)	0.845 **
	No	18 (26.09)	51 (73.91)		14 (20)	56 (80)	
Hypoxia	Yes	1 (16.67)	5 (83.33)	0.846	1 (16.67)	5 (83.33)	1 **
	No	12 (20)	48 (80)		9 (14.75)	52 (85.25)	
Blood Glucose (mg/dL)	70–200	8 (20.51)	31 (79.49)	1	6 (15)	34 (85)	1 ***
	>200	0 (0)	3 (100)		0 (0)	3 (100)	
	<70	0 (0)	1 (100)		0 (0)	1 (100)	
Hypoglycemia	Yes	1 (20)	4 (80)	0.79	1 (20)	4 (80)	0.975 **
	No	17 (25.37)	50 (74.63)		14 (20.59)	54 (79.41)	
Hyperglycemia	Yes	10 (26.32)	28 (73.68)	0.734	8 (21.05)	30 (78.95)	0.864 **
	No	8 (22.86)	27 (77.14)		7 (19.44)	29 (80.56)	
number of brain radiographic injuries	≥2	17 (34.69)	32 (65.31)	0.011	15 (30)	35 (70)	0.001 **
	1	2 (7.69)	24 (92.31)		0 (0)	26 (100)	
Neurosurgical procedure	Yes	12 (32.43)	25 (67.57)	0.166	10 (27.03)	27 (72.97)	0.122 **
	No	7 (18.42)	31 (81.58)		5 (12.82)	34 (87.18)	
Infection	Yes	13 (34.21)	25 (65.79)	0.05	11 (28.95)	27 (71.05)	0.058 **
	No	5 (14.29)	30 (85.71)		4 (11.11)	32 (88.89)	
Intracranial Hypertension	Yes	4 (33.33)	8 (66.67)	0.388	4 (33.33)	8 (66.67)	0.176 **
	No	13 (21.67)	47 (78.33)		10 (16.39)	51 (83.61)	

Bpm—beats per minute; GCS–Glasgow Coma Scale; ICU—Intensive Care Unit; ISS—Injury Severity Score; LOS—Length of stay; Max—Maximum value; mg/dL—milligrams per deciliter; Min—Minimum value; RR—respiratory rate; SD—Standard deviation. * Wilcoxon-Mann–Whitney; ** Pearson’s chi-squared test; *** Fisher test; **** Brunner-Munze test.

**Table 6 jcm-11-06466-t006:** Mortality and dependence based on TBI lesion.

Lesion		Mortality in 14 Days			Dependence in 6 Months			Dependence in 12 Months		
		Yes	No	*p*-Value	Yes	No	*p*-Value	Yes	No	*p*-Value
Epidural Hematoma (EDH)	Yes	14	27	0.494 *	4	20	0.236 *	4	20	0.763 **
	No	26	66		15	36		11	41	
Focal lesion without EDH	Yes	26	66	0.494 *	15	36	0.271 *	11	41	0.763 **
	No	14	27		4	20		4	20	
Subdural Hematoma (SDH)	Yes	20	42	0.608 *	11	21	0.120 *	9	24	0.148 *
	No	20	51		8	35		6	37	
Contusion	Yes	6	24	0.171 *	4	15	0.765 **	2	17	0.330 **
	No	34	69		15	41		13	44	

Focal lesion without EDH: patient with contusion and SDH. * Pearson’s chi-squared test; ** Fisher test.

## Data Availability

The data presented in this study are available within this article and Appendix A.

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
