# Peer review of "Long-Term Outcomes in Severe Traumatic Brain Injury and Associated Factors: A Prospective Cohort Study"

_jcm, 2022, doi:10.3390/jcm11216466_

Round 1
Reviewer 1 Report
There are a few aspect that should be improved.
Please see the comments in the attachment.

Author Response
REVIEWER 1:
Add the global burden statistics for TBI and perhaps a statement of the burden of TBI in Latin America
We added this information to our introduction
Define severe TBI here as admission GCS < =8)
We added this information
What does this sentence indicate? tSAH included or excluded, please clarify.
Patients with focal lesion and tSAH were included. Exclusive tSAH were excluded. We added a statement clarifying
Please provide characteristics of your hospital. Is it a level I trauma center? What is the bed capacity? Where are patients with TBI cared for? Dedicated neurocritical care unit?
We added this information
Not clear what this means. Do you mean polytrauma not included?
Do you mean isolated TBI not undergoing neurosurgical interventions were excluded?
Patients with polytrauma were included. Isolated TBI not undergoing interventions but with abnormal CT were included. We clarified this information in our article.
Define hypotension
We included the definition of hypotension
define blood glucose targets used at your hospital
We included the target range
what type of infection, hospital-acquired? CLABSI, CAUTI. VAP, C.DIFF? please clarify.
All those conditions were included. We clarified in our text that any condition that required medical management were included.
Provide statement as to who performed these assessments? Were these blinded-assesor? or done by someone from the treatment team?
We included a statement that all analysis were blindly performed by the author CPES.
I am assuming you performed Shapiro-Wilk test to examine normality of distribution of continuous variables? Please include normality testing here.
We included the normality assessment
Can you comment on what R packages were used for analysis?
The following packages were used: tidyverse, ggpubr, rstatix, readxl. We included this in our article now
Define abnormality in RR, is it tachypnea > 35 or bradypnea or apnea?
Normal range of RR was defined as 12 – 20
Define HR abnormal values. is it tachycardia or bradycardia?
Normal range of HR was defined as 50 - 100
Define hypoxia.
(SpO2 < 92%)
define pupillary changes, is it non-reactive pupils? or anisocoria?
Both non reactivity and anisocoria were considered.
make sure you include SAH in methods earlier
We corrected this
this is not defined in methods. what doe this mean? is it time on mechanical ventilation with sedation? was every patient mechanically ventilated?
Mean sedation is the time on mechanical ventilation with sedation. All patients that were intubated were mechanically ventilated. We included this definition in our methodology.
this is not defined in methods. how was intracranial HTN defined?
HTN was defined by clinical examination findings and CT imaging. We included this in our methodology now.
grammatically incorrect. you can say underwent a neurosurgical procedure. Define what that was.
We corrected this.
this was not defined earlier in methods.
We included our variables not defined previously in our methodology now.
spelling mistake, please correct.
We corrected this mistake
i am not sure if the number is important of type of lesion is. i would recommend adding type of lesion in your univariate model.
The comparison of each type of lesion and their outcome can be found in table 6

Reviewer 2 Report
General comments:
Predicting recovery in patients with TBI is a difficult but important task. The authors have conducted a cohort study to evaluate patterns of functional recovery, ADL, and IADL in patients with severe TBI and focal injury. Although the topic is important, I find the study to lack merit and methodological quality. I have listed my major and minor concerns below.
Particularly, I would recommend the authors seek statistical guidance. I will not recommend publication in its current form and suggest a thorough and major revision before resubmitting this manuscript.
Major concerns:
Protocol and adherence to STROBE. Although not always considered mandatory in observational studies, I would like to emphasise the advantages of using a pre-published protocol in observational studies. With more than 20 outcome variables it is difficult to see this study as anything but a data mining project. A protocol would supply the reader with clear definitions of which of these are considered outcomes, exposures, predictors, potential confounders, or effect modifiers. The authors should also follow the STROBE statement (https://www.strobe-statement.org/).
I find the aim of the study (lines 75 to 81) imprecise. This reflects the conclusion which seems to be more or less a repetition of some of the results.
The method section 2.2 is confusing to read. It should be divided into outcomes, exposures, predictors, potential confounders, and effect modifiers.
The authors make several comparisons between two groups (14-day mortality yes-no; 6-months disability yes-no; or 12-months disability yes-no). Furthermore, all outcomes are compared over at least three to four periods. This reviewer has found at least 63 comparisons increasing the risk of multiplicity issues substantially.
The authors present in the method section that they will use a variety of statistical methods. Yet none of these is specified in the actual analysis in tables 2-4. This impairs the transparency of the results and the reader is left without any chance of knowing what actual analysis method was used.
In figure 1 patients who have died are excluded at other time points. This cannot be the correct way to do this. It is always difficult to handle missing information but to ignore it is incorrect. The authors should present all data with intention to treat analysis.
The presentation of most of the observed results is confusing. It is not clear what the denominator is! Please be concise in this. Furthermore, there is no need to present proportions with decimals.
Minor concerns:
The authors should prof read their text more thoroughly. Please use a period as a decimal separator and not a comma.
Line 60: “…some studies…” does not seem to be referenced.
Line 70: IALD should be IADL.
Line 110-111: “Presence of infections”, “presence of intracranial hypertension” or ”neurosurgical treatment” how do the authors define this?
Line 120: The sentence does not seem to be finished.
Line 120-124: This section is incoherent. Different outcome measures are mentioned and the authors refer to “this instrument” (GOS-E???), but GOS-E is hardly used to detect subtle changes.
Line 139-153: This section needs more details. How are data in general presented? How is missing data handled? Specifically, for each statistical comparison, how is this handled? Please elaborate. As mentioned above this reviewer believes that a significance level of 5% is naïve.
Line 157: “133 met the inclusion and exclusion criteria.” They could not have met the exclusion criteria!
Section 3.1: Data in brackets are standard deviations? It is not clear
Line 161-162: “Of the 82 surviving patients, 75 patients (56.4%) participated in the entire follow-161 up and 7 (5.3%) participated only in part of the follow-up (discharge).” So of 75 of 82 patients is 56%?
Line 163: “most individuals (68%)…” what does this refer to. What is the denominator? Please check the remaining manuscript for similar flaws.
Line 169: “GCS score =3” rephrase to GCS score of three.
Line 173: what are abnormal values?
Line 178: how is it possible to have more than one brain injury? Do the authors mean that there is more than one injury visible on the CTor MR scan? Or that patients had a prior injury?
Line 183: “with a mean of 53, 27 (sd = 15.212) and…”. What do the authors mean by “53, 27” and what does “sd” mean?
Line 187: how can you be admitted for zero days?
Most of the data in section 3.1 should be presented in a table
Line 199: “…51 patients (38.35%) had died…” This has already been presented above.
Line 210-211: “…(7, 9%) had persistent vegetative (Figure 1).” You cannot have persistent vegetative.
Figure 1: Numbers are hard to read. As mentioned above patients have still died at 3, 6, and 12 months.
Author Response
General comments:
Predicting recovery in patients with TBI is a difficult but important task. The authors have conducted a cohort study to evaluate patterns of functional recovery, ADL, and IADL in patients with severe TBI and focal injury. Although the topic is important, I find the study to lack merit and methodological quality. I have listed my major and minor concerns below.
Particularly, I would recommend the authors seek statistical guidance. I will not recommend publication in its current form and suggest a thorough and major revision before resubmitting this manuscript.
Major concerns:
Protocol and adherence to STROBE. Although not always considered mandatory in observational studies, I would like to emphasise the advantages of using a pre-published protocol in observational studies. With more than 20 outcome variables it is difficult to see this study as anything but a data mining project. A protocol would supply the reader with clear definitions of which of these are considered outcomes, exposures, predictors, potential confounders, or effect modifiers. The authors should also follow the STROBE statement (https://www.strobe-statement.org/).
Dear reviewer, we agree with the importance of pre published protocol in observational studies. Unfortunately, our protocol was not public published, but only submitted to the ethics committee. We followed all pre-determined analysis. The large number of outcomes are merely similar ways to widely described in the literature to measure the same outcome (mortality and morbidity). As shown in our article, they were highly correlated. The necessary statistical adjustments for multiple comparison where used when necessary. We changed our article to follow the STROBE statement.
I find the aim of the study (lines 75 to 81) imprecise. This reflects the conclusion which seems to be more or less a repetition of some of the results.
We changed the way we presented our objective so it is concise and precise.
The method section 2.2 is confusing to read. It should be divided into outcomes, exposures, predictors, potential confounders, and effect modifiers.
We added this division to our methodology
The authors make several comparisons between two groups (14-day mortality yes-no; 6-months disability yes-no; or 12-months disability yes-no). Furthermore, all outcomes are compared over at least three to four periods. This reviewer has found at least 63 comparisons increasing the risk of multiplicity issues substantially.
The appropriate statistical corrections for multiple comparison were used when indicated (Holm–Bonferroni method). We included this information in our methodology.
The authors present in the method section that they will use a variety of statistical methods. Yet none of these is specified in the actual analysis in tables 2-4. This impairs the transparency of the results and the reader is left without any chance of knowing what actual analysis method was used.
We included in our tables the test used for each analysis now
In figure 1 patients who have died are excluded at other time points. This cannot be the correct way to do this. It is always difficult to handle missing information but to ignore it is incorrect. The authors should present all data with intention to treat analysis.
Thank you very much for this observation, we corrected figure 1 including patients that died.
The presentation of most of the observed results is confusing. It is not clear what the denominator is! Please be concise in this. Furthermore, there is no need to present proportions with decimals.
We corrected this, a new table with most of the information was added to our article. Since some variables were not available for all patients, we include the percentage after each value. We excluded the decimals for the proportions
Minor concerns:
The authors should prof read their text more thoroughly. Please use a period as a decimal separator and not a comma.
Line 60: “…some studies…” does not seem to be referenced.
Line 70: IALD should be IADL.
Line 110-111: “Presence of infections”, “presence of intracranial hypertension” or ”neurosurgical treatment” how do the authors define this?
Line 120: The sentence does not seem to be finished.
Line 120-124: This section is incoherent. Different outcome measures are mentioned and the authors refer to “this instrument” (GOS-E???), but GOS-E is hardly used to detect subtle changes.
Line 139-153: This section needs more details. How are data in general presented? How is missing data handled? Specifically, for each statistical comparison, how is this handled? Please elaborate. As mentioned above this reviewer believes that a significance level of 5% is naïve.
Line 157: “133 met the inclusion and exclusion criteria.” They could not have met the exclusion criteria!
Section 3.1: Data in brackets are standard deviations? It is not clear
Line 161-162: “Of the 82 surviving patients, 75 patients (56.4%) participated in the entire follow-161 up and 7 (5.3%) participated only in part of the follow-up (discharge).” So of 75 of 82 patients is 56%?
Line 163: “most individuals (68%)…” what does this refer to. What is the denominator? Please check the remaining manuscript for similar flaws.
Line 169: “GCS score =3” rephrase to GCS score of three.
Line 173: what are abnormal values?
Line 178: how is it possible to have more than one brain injury? Do the authors mean that there is more than one injury visible on the CTor MR scan? Or that patients had a prior injury?
Line 183: “with a mean of 53, 27 (sd = 15.212) and…”. What do the authors mean by “53, 27” and what does “sd” mean?
Line 187: how can you be admitted for zero days?
Most of the data in section 3.1 should be presented in a table
Line 199: “…51 patients (38.35%) had died…” This has already been presented above.
Line 210-211: “…(7, 9%) had persistent vegetative (Figure 1).” You cannot have persistent vegetative.
Figure 1: Numbers are hard to read. As mentioned above patients have still died at 3, 6, and 12 months.
We were able to correct all minors’ concerns. Thank you very much for this detailed review.

Reviewer 3 Report
Comment Section:
General comment |
The objective of the study was to evaluate the pattern of functional recovery, ADL, and IADL of victims of severe TBI with focal injury up to twelve months after trauma and factors associated with mortality and recovery at six and twelve months after the trauma. The subject is interesting and it explores one of the areas of neurointensive care that concerns a major public health problem in many countries around the world, such as TBI. The study provides an attempt to understand the evolution of patients with TBI and focal injury, relating various predictors of evolution. However, there are some important limitations to consider. It would be advisable to make the following modifications classified, in my opinion, in major (“very important or almost mandatory) and minor (“suggested”).
|
Introduction |
Major comments: Line 75 to 81, It is mandatory that this last paragraph of the introduction be placed before the previous paragraph. Because the objectives of the study should be the last thing detailed in the introduction. Therefore, move the last paragraph of the introduction so that it comes before the paragraph that describes the objective of the study. Minor comments: I would try to add more references from the last 5 years in the introduction. Since only 5 (28%) of the 18 references in the introduction are from the last 5 years.
|
Methods |
Major comments: Line 83, I consider it very important to add the subsections within Methods: Design, Population, Inclusion and exclusion criteria, Sampling/Size, Interventions, Definitions, Variables, Statistical analysis, Ethical aspects. Although this is not something fundamental, I believe that it allows whoever reads the manuscript to follow an order in the methodological characteristics of the work that makes it much easier to read and understand. I think this would greatly improve the order in which Methods aspects should be described. Line 84, Here the observational characteristic should be added to the study design, leaving: “This is an observational prospective cohort study…”. Line 139, Statistical Analysis: This section should clarify how the various variables were expressed (which are later shown in results). For example: “The continuous data with a normal distribution are expressed as means and SDs, whereas the non-normally distributed data are presented as medians with IQRs. Categorical variables are expressed as frequencies and percentages...” Minor comments: Line 100, It would be convenient to add here that it was carried out following the recommendations of the Declaration of Helsinki II (World Medical Association Declaration of Helsinki: ethical principles for medical research involving human subjects. J Am Med Assoc. 2013 Nov 27;310(20):2191,4).
|
Results |
Major comments: Line 170, In the results section, the existing discrepancy can be observed with the lack of clarification of the management of the variables in the Statistical analysis section. All this should be homogenized, since, for example, in line 158 they say that the average age, in my interpretation, was 34.08 with a supposed SD of 12.35 years (since they do not clarify it), and in line 170 it says GCS of 4.30 (SD 1.82). That is to say, in one place they clarify that it is SD and in another place it is not. Please clarify in the Statistical analysis section and homogenize the results. Line 212, Please improve Figure 1, since the percentages cannot be seen clearly. Line 235, Please correct Portuguese words from Table 2! Like "Tempo".Minor comments: Line 156, Here I would add a figure with a flowchart on sample selection. Line 187, If you use SD, do not write it in lowercase. The same applies to line 194 and in successive places within results. |
Discussion |
Major comments: Line 414, I think that in this section of limitations, an important limitation of the work should be added, which refers to the lack of data on the rehabilitation of patients after hospital discharge. Point that could have had an important interference and its analysis would have been very useful. Although it is clarified that many of the patients did not have access to said adequate rehabilitation, this point that I mention must be clarified.Minor comments: Line 391, I believe that adding a reference to a Latin American paper on the subject would enrich this part of the discussion, since it would be compared with regional data. |
Conclusions |
Major comments: Line 423, Although I consider that, in general, the conclusion is adequate, I think that the characteristics of the center where the study was carried out should be added. Also, I think a few words should be added about the usefulness of the results of the work for clinical practice.Minor comments: None. |
References |
Major comments: All references must follow the guide of authors of the journal. Moreover, the previous suggestion about local papers (Latin America), should be done.Minor comments: None. |
Other Comments |
Title: I think it is right.
|
Additional comments:
Abstract:
The abstract should be adapted to the major modifications suggested in the manuscript. In addition, the acronym TBI should be clarified the first time it is used.
Keywords:
To make the keywords appear in MeSH, modifications must be made. The keyword “Glasgow outcome scale extended” should be changed for “Glasgow outcome scale”, because it is the way it appears in the MeSH.
The keyword “Recovery” should be changed for “Recovery of Function”.
Ethical review:
It needs to be added the point of the Declaration of Helsinki.
Author Response
General comment
The objective of the study was to evaluate the pattern of functional recovery, ADL, and IADL of victims of severe TBI with focal injury up to twelve months after trauma and factors associated with mortality and recovery at six and twelve months after the trauma.
The subject is interesting and it explores one of the areas of neurointensive care that concerns a major public health problem in many countries around the world, such as TBI. The study provides an attempt to understand the evolution of patients with TBI and focal injury, relating various predictors of evolution. However, there are some important limitations to consider.
It would be advisable to make the following modifications classified, in my opinion, in major (“very important or almost mandatory) and minor (“suggested”).
Introduction
Major comments:
Line 75 to 81, It is mandatory that this last paragraph of the introduction be placed before the previous paragraph. Because the objectives of the study should be the last thing detailed in the introduction. Therefore, move the last paragraph of the introduction so that it comes before the paragraph that describes the objective of the study.
We changed the order of this paragraph, so the objective is in the last paragraph of the introduction.
Minor comments:
I would try to add more references from the last 5 years in the introduction. Since only 5 (28%) of the 18 references in the introduction are from the last 5 years.
We added three recent articles to our introduction.
Methods
Major comments:
Line 83, I consider it very important to add the subsections within Methods: Design, Population, Inclusion and exclusion criteria, Sampling/Size, Interventions, Definitions, Variables, Statistical analysis, Ethical aspects. Although this is not something fundamental, I believe that it allows whoever reads the manuscript to follow an order in the methodological characteristics of the work that makes it much easier to read and understand. I think this would greatly improve the order in which Methods aspects should be described.
We divided according to your suggestion
Line 84, Here the observational characteristic should be added to the study design, leaving: “This is an observational prospective cohort study…”.
We added this information
Line 139, Statistical Analysis: This section should clarify how the various variables were expressed (which are later shown in results). For example: “The continuous data with a normal distribution are expressed as means and SDs, whereas the non-normally distributed data are presented as medians with IQRs. Categorical variables are expressed as frequencies and percentages...”
We included this in our article
Minor comments:
Line 100, It would be convenient to add here that it was carried out following the recommendations of the Declaration of Helsinki II (World Medical Association Declaration of Helsinki: ethical principles for medical research involving human subjects. J Am Med Assoc. 2013 Nov 27;310(20):2191,4).
We added this in the 2.5. Ethical aspects section.
Results
Major comments:
Line 170, In the results section, the existing discrepancy can be observed with the lack of clarification of the management of the variables in the Statistical analysis section. All this should be homogenized, since, for example, in line 158 they say that the average age, in my interpretation, was 34.08 with a supposed SD of 12.35 years (since they do not clarify it), and in line 170 it says GCS of 4.30 (SD 1.82). That is to say, in one place they clarify that it is SD and in another place it is not. Please clarify in the Statistical analysis section and homogenize the results.
We include a statement in the statistical analysis section and homogenize it through the text.
Line 212, Please improve Figure 1, since the percentages cannot be seen clearly.
We included a new Figure 1, with data for all patients and with a better layout.
Line 235, Please correct Portuguese words from Table 2! Like "Tempo".
We corrected this and other mistakes
Minor comments:
Line 156, Here I would add a figure with a flowchart on sample selection.
We included a new figure with this flowchart
Line 187, If you use SD, do not write it in lowercase. The same applies to line 194 and in successive places within results.
We corrected this
Discussion
Major comments:
Line 414, I think that in this section of limitations, an important limitation of the work should be added, which refers to the lack of data on the rehabilitation of patients after hospital discharge. Point that could have had an important interference and its analysis would have been very useful. Although it is clarified that many of the patients did not have access to said adequate rehabilitation, this point that I mention must be clarified.
We added this limitation to our discussion
Minor comments:
Line 391, I believe that adding a reference to a Latin American paper on the subject would enrich this part of the discussion, since it would be compared with regional data.
We added an article with data for Latin America
Conclusions
Major comments:
Line 423, Although I consider that, in general, the conclusion is adequate, I think that the characteristics of the center where the study was carried out should be added. Also, I think a few words should be added about the usefulness of the results of the work for clinical practice.
We added this to our conclusion
Minor comments:
None.
References
Major comments:
All references must follow the guide of authors of the journal. Moreover, the previous suggestion about local papers (Latin America), should be done.
We corrected the format for all references and included new articles
Minor comments:
None.
Other Comments
Title:
I think it is right.
Additional comments:
Abstract:
The abstract should be adapted to the major modifications suggested in the manuscript. In addition, the acronym TBI should be clarified the first time it is used.
We corrected this
Keywords:
To make the keywords appear in MeSH, modifications must be made. The keyword “Glasgow outcome scale extended” should be changed for “Glasgow outcome scale”, because it is the way it appears in the MeSH.
The keyword “Recovery” should be changed for “Recovery of Function”.
We changed those keywords
Ethical review:
It needs to be added the point of the Declaration of Helsinki.
We added this to our methodology

Reviewer 4 Report
Viera de Oliveira et al. describe the patterns of recovery for patients who have a focal brain lesion after severe traumatic brain injury in a prospective databased in a developing nation. The authors have a comprehensive set of variables collected during initial admission including radiographic patterns of injury and treatment in hospital. They record outcomes at discharge, 3-, 6-, and 12-months post-trauma using several different scales of functional independence including the GOSE, Disability Rating Scale, Katz Indez, and Lawton scale. The authors find that the fastest recovery after severe TBI occurs from discharge to 3 months. Interestingly, they do not find significant recovery from 6- to 12-months post-trauma. Using simple descriptive statistics, the authors find several characteristics that correlate with worse functional outcomes.
The authors should be commended for developing the most comprehensive and complete database this reviewer has seen, regardless of location. Their follow up rate of 126/133=~95% exceeds the norm in clinical trials in developed nations.1 In addition, the authors have multiple follow up scales that provide unique insights into the recovery of TBI, including the relative recovery of different functional domains. This dataset has the potential to provide numerous insights into the functional recovery of severe TBI patients.
Major Comments:
1) While the authors have developed a strong outcomes database, this manuscript lacks a clear purpose or hypothesis. The authors initially state their goal is to assess the recovery of patients with focal brain lesions, but do not spend most of the manuscript exploring this topic. The authors veer off topic on several occasions. This manuscript would best be served through a significant revision that focuses on one topic and provides evidence to draw conclusions from their data. The authors should clearly state their hypothesis at the end of the introduction.
2) The authors do not cite or discuss several of the key papers in TBI literature that are directly relevant to their topic. Yu et al. describe the patterns of CT findings and their relationship to outcomes in TRACK-TBI and validate this in CENTRE-TBI. This is the most comprehensive analysis of the relationship between outcomes and cranial imaging findings. The authors could emphasize how their work focuses on developing nations, include TRACK-TBI or Centre-TBI. Additionally, the authors do not cite how their patterns of recovery are similar or different than several recent papers looking at functional outcomes after TBI.2,3
3) The analysis approach fails to capture the repeated measures design of their data. The authors only use basic descriptive statistics to find correlations in their data. As stands, I do not know the relative importance of the features associated with outcomes. Multivariate modeling, especially multivariate modeling that captures the repeated measures design of their data, would allow for more meaningful conclusions.
4) The authors fail to clearly define what a focal brain lesion is. The state that subarachnoid hemorrhage should be excluded, but do not discuss patients who have small subdurals with otherwise diffuse brain lesions. Bunching all types of focal lesions together is probably not the best approach. Epidural hematomas are known to be associated with positive outcomes while other types are not. Lumping a positive predictor in with negative predictors weakens associations.4
Minor comments:
1) TBI is THE leading cause of death and disability in <40 worldwide, not a leading cause in select countries.
2) This statement is not clear; I do not understand it: “Participants diagnosed with severe TBI with focal lesions, 92 associated or not with other types of TBI and undergoing neurosurgical procedures or 93 not, were included in the research.”
3) Why exclude spinal cord lesions? A brain specific GOSE scale exists for this purpose. If the authors do not have that recorded, it is fair to exclude spinal cord injuries, but should better justify this reason.
4) The atuhors do not discuss how their data was collected. Were all measurements prospectively recorded, or most retrospectively collected? For example, was GCS prospectively collected in real time or retrospectively collected form chart reviews?
5) Only 1/3 of patients were intubated. Is it standard of care to intubate patients with GCS 8 or less in trauma. Why a discordance?
6) Many newer clinical trials are using GOSE 4-8 as favorable. Maybe perform a sub-analysis with a different cutoff to see if your results hold?
7) The authors should follow some guideline, such as one for epidemiology studies. (https://www.equator-network.org/)
8) Should display CONSORT diagram
9) What do the numbers in parenthesis, such as 12 after age mean? Is this the standard deviation? Sometimes you define this and put SD =7.16…… Other times it is randomly presented as information. Additionally, the authors change the number of decimal points used throughout the manuscript. Sometimes numbers are 38.01…verses 12.8%....versus 68%. The atuhors should pick a style and be consistent.
10) What does pupillary changes mean? This should be clearly defined
11) What does heart rate abnormalities mean? Is this bradycardia or tachycardia?
12) Describing an injury as multiple brain injuries is atypical. It often is described as multiple CT findings or radiographic injury findings. Your descriptions suggests there are more than one injury.
12) 3% of patients with severe TBi were not admitted to an ICU. This need to be explained as this is a deviation of standard of care in most countries.
13) How did you measure intracranial hypertension? Do you standardly place ICP monitors?
149) Figure 1 cuts of numbers in the figure
14) Their domain analysis is super interesting and very very unusual. Very few groups have this type of data set, especially in developing nations. A focus on the deficiencies of recovery in subdomains in TBI is a paper in itself.
15) You did not find a difference between 6 and 12 months post trauma, yet developing nations do find a difference over this time point. This should be explored or explained more.
16) The relationship between sedation time and survival is confounded because people who die early cannot have prolonged sedation.
17) The lack of relationship between EDH and functional outcomes needs to be explained because this has repeatedly been found to predict good outcomes in multiple studies
18) You stated that the main cause of death was brain death 55% and cardiopulmonary arrest ~25%. This is the not true in developed nations, where withdrawal of care is the primary cause of death in TBI. Was formal brain death tested performed? This finding is presented in the discussion (should be in results) and should be explored significantly.
19) The authors do not meaningfully compare the various scales besides superficially stating that they are different.
Works Cited
1. Richter S, Stevenson S, Newman T, et al. Handling of missing outcome data in traumatic brain injury research: A systematic review. J Neurotrauma. 2019;36(19):2743-2752. doi:10.1089/neu.2018.6216
2. Kowalski RG, Hammond FM, Weintraub A, et al. Recovery of Consciousness and Functional Outcome in Moderate and Severe Traumatic Brain Injury. JAMA Neurol. 2021;In press:1-10. doi:10.1001/jamaneurol.2021.0084
3. McCrea MA, Giacino JT, Barber J, et al. Functional Outcomes over the First Year after Moderate to Severe Traumatic Brain Injury in the Prospective, Longitudinal TRACK-TBI Study. JAMA Neurol. 2021;78(8):982-992. doi:10.1001/jamaneurol.2021.2043
4. Steyerberg EW, Mushkudiani N, Perel P, et al. Predicting outcome after traumatic brain injury: Development and international validation of prognostic scores based on admission characteristics. PLoS Med. 2008;5(8):1251-1261. doi:10.1371/journal.pmed.0050165
Author Response
Viera de Oliveira et al. describe the patterns of recovery for patients who have a focal brain lesion after severe traumatic brain injury in a prospective databased in a developing nation. The authors have a comprehensive set of variables collected during initial admission including radiographic patterns of injury and treatment in hospital. They record outcomes at discharge, 3-, 6-, and 12-months post-trauma using several different scales of functional independence including the GOSE, Disability Rating Scale, Katz Indez, and Lawton scale. The authors find that the fastest recovery after severe TBI occurs from discharge to 3 months. Interestingly, they do not find significant recovery from 6- to 12-months post-trauma. Using simple descriptive statistics, the authors find several characteristics that correlate with worse functional outcomes.
The authors should be commended for developing the most comprehensive and complete database this reviewer has seen, regardless of location. Their follow up rate of 126/133=~95% exceeds the norm in clinical trials in developed nations.1 In addition, the authors have multiple follow up scales that provide unique insights into the recovery of TBI, including the relative recovery of different functional domains. This dataset has the potential to provide numerous insights into the functional recovery of severe TBI patients.
Major Comments:
1) While the authors have developed a strong outcomes database, this manuscript lacks a clear purpose or hypothesis. The authors initially state their goal is to assess the recovery of patients with focal brain lesions, but do not spend most of the manuscript exploring this topic. The authors veer off topic on several occasions. This manuscript would best be served through a significant revision that focuses on one topic and provides evidence to draw conclusions from their data. The authors should clearly state their hypothesis at the end of the introduction.
Thank you very much for this observation. Indeed, our objective was not clear. The focus of our article was to evaluate factors in the initial patient care that might be associated with better or worse outcomes after TBI. We changed this in our introduction and try to focus the rest of the article for this objective. We also added in our limitation the lack of data regarding detailed information for what kind of rehabilitation each patient had and how this could impact in their outcome.
2) The authors do not cite or discuss several of the key papers in TBI literature that are directly relevant to their topic. Yu et al. describe the patterns of CT findings and their relationship to outcomes in TRACK-TBI and validate this in CENTRE-TBI. This is the most comprehensive analysis of the relationship between outcomes and cranial imaging findings. The authors could emphasize how their work focuses on developing nations, include TRACK-TBI or Centre-TBI. Additionally, the authors do not cite how their patterns of recovery are similar or different than several recent papers looking at functional outcomes after TBI.2,3
We included all suggested articles in our discussion. We also reviewed the literature for new articles in the subject and included in our article.
3) The analysis approach fails to capture the repeated measures design of their data. The authors only use basic descriptive statistics to find correlations in their data. As stands, I do not know the relative importance of the features associated with outcomes. Multivariate modeling, especially multivariate modeling that captures the repeated measures design of their data, would allow for more meaningful conclusions.
We performed a multivariate model, using automated stepwise selection for the most important variables to include in our model. The following variables were included: Age, more than one radiographic lesion on CT imaging, length of stay in ICU, ISS at admission and GCS at admission. We also analyzed if there were a correlation between each of those variables and change over time using repeated measures with generalized estimating equations.
We included our final model in the supplementary material:
Variable |
Estimate |
p value for model |
p -value for change over time |
Intercept |
-0.064 |
||
Age |
0.113 |
0.500 |
0.439 |
ICU LOS |
-0.045 |
0.016 |
0.154 |
More than one brain lesion |
-0.592 |
0.149 |
0.065 |
ISS |
0.045 |
0.064 |
0.127 |
GCS |
-0.334 |
0.008 |
0.307 |
4) The authors fail to clearly define what a focal brain lesion is. The state that subarachnoid hemorrhage should be excluded, but do not discuss patients who have small subdurals with otherwise diffuse brain lesions. Bunching all types of focal lesions together is probably not the best approach. Epidural hematomas are known to be associated with positive outcomes while other types are not. Lumping a positive predictor in with negative predictors weakens associations.4
We included a more detailed description of which lesions were included in our article. Table 6 shows outcomes based on each type of lesion. We also included in our discussion that specific aspects of each type of lesion can have an impact in their outcome
Minor comments:
1) TBI is THE leading cause of death and disability in <40 worldwide, not a leading cause in select countries.
We corrected this
2) This statement is not clear; I do not understand it: “Participants diagnosed with severe TBI with focal lesions, 92 associated or not with other types of TBI and undergoing neurosurgical procedures or 93 not, were included in the research.”
We simplified this statement and included the necessary information in the correct part of the article
3) Why exclude spinal cord lesions? A brain specific GOSE scale exists for this purpose. If the authors do not have that recorded, it is fair to exclude spinal cord injuries, but should better justify this reason.
We included this limitation in our discussion
4) The atuhors do not discuss how their data was collected. Were all measurements prospectively recorded, or most retrospectively collected? For example, was GCS prospectively collected in real time or retrospectively collected form chart reviews?
Data was prospectively collected. We included this statement in our methodology
5) Only 1/3 of patients were intubated. Is it standard of care to intubate patients with GCS 8 or less in trauma. Why a discordance?
Clinical decision for intubation was based on individual factors at admission. GCS scale was one of the factors used for the decision, but others such as oxygen saturation, ability to maintain their airway patency and type of injury were also considered.
6) Many newer clinical trials are using GOSE 4-8 as favorable. Maybe perform a sub-analysis with a different cutoff to see if your results hold?
Following the recommendation of the other reviewer, we tried to reduce the number of comparisons in our article to avoid bias. For that reason, we did not include this comparison in this new version.
7) The authors should follow some guideline, such as one for epidemiology studies. (https://www.equator-network.org/)
We adapted our article to follow STROBE guidelines
8) Should display CONSORT diagram
We added a new figure (FIGURE 1) for this
9) What do the numbers in parenthesis, such as 12 after age mean? Is this the standard deviation? Sometimes you define this and put SD =7.16…… Other times it is randomly presented as information. Additionally, the authors change the number of decimal points used throughout the manuscript. Sometimes numbers are 38.01…verses 12.8%....versus 68%. The atuhors should pick a style and be consistent.
We changed this in our article now. We detailed how data is presented in our methodology and included the standard deviation (SD x) in our results for all numbers.
10) What does pupillary changes mean? This should be clearly defined
We defined this in our methodology now
11) What does heart rate abnormalities mean? Is this bradycardia or tachycardia?
We defined this in our results now
12) Describing an injury as multiple brain injuries is atypical. It often is described as multiple CT findings or radiographic injury findings. Your descriptions suggests there are more than one injury.
We corrected this
12) 3% of patients with severe TBi were not admitted to an ICU. This need to be explained as this is a deviation of standard of care in most countries.
Those patients were not in the ICU due to lack of resources and space available in the ICU, but were properly monitored as they would in the ICU. We added this to our article
13) How did you measure intracranial hypertension? Do you standardly place ICP monitors?
149) Figure 1 cuts of numbers in the figure
We defined intracranial hypertension in our methodology now. ICP monitors is not standardly use for every patient, but rather decided in an individual basis. We corrected figure 1.
14) Their domain analysis is super interesting and very very unusual. Very few groups have this type of data set, especially in developing nations. A focus on the deficiencies of recovery in subdomains in TBI is a paper in itself.
Thank you
15) You did not find a difference between 6 and 12 months post trauma, yet developing nations do find a difference over this time point. This should be explored or explained more.
We explained in our discussion now that the lack of improvement is probably due to problems to access to a rehabilitation program. This probably has a direct impact in the outcome of those patients
16) The relationship between sedation time and survival is confounded because people who die early cannot have prolonged sedation.
That’s completely true. We included this intrinsic limitation to our discussion.
17) The lack of relationship between EDH and functional outcomes needs to be explained because this has repeatedly been found to predict good outcomes in multiple studies
We included this in our discussion, showing that other studies found this correlation. The presence of other concomitant injuries and unknow rehabilitation program might have impacted in our results.
18) You stated that the main cause of death was brain death 55% and cardiopulmonary arrest ~25%. This is the not true in developed nations, where withdrawal of care is the primary cause of death in TBI. Was formal brain death tested performed? This finding is presented in the discussion (should be in results) and should be explored significantly.
All patients had formal brain death tested performed, according to local legislation. We included this in our results and discussed in the article, including new references.
19) The authors do not meaningfully compare the various scales besides superficially stating that they are different.
The focus of the article was not to compare the differences between the scales or decide if one of them was better than the other for TBI. They can show slightly different information. We added this information to our article so the reader can use the scale that he is most familiar with.
Below you can find a sensitivity and specificity of each scale for 14 days mortality.
Works Cited
- Richter S, Stevenson S, Newman T, et al. Handling of missing outcome data in traumatic brain injury research: A systematic review. J Neurotrauma. 2019;36(19):2743-2752. doi:10.1089/neu.2018.6216
- Kowalski RG, Hammond FM, Weintraub A, et al. Recovery of Consciousness and Functional Outcome in Moderate and Severe Traumatic Brain Injury. JAMA Neurol. 2021;In press:1-10. doi:10.1001/jamaneurol.2021.0084
- McCrea MA, Giacino JT, Barber J, et al. Functional Outcomes over the First Year after Moderate to Severe Traumatic Brain Injury in the Prospective, Longitudinal TRACK-TBI Study. JAMA Neurol. 2021;78(8):982-992. doi:10.1001/jamaneurol.2021.2043
- Steyerberg EW, Mushkudiani N, Perel P, et al. Predicting outcome after traumatic brain injury: Development and international validation of prognostic scores based on admission characteristics. PLoS Med. 2008;5(8):1251-1261. doi:10.1371/journal.pmed.0050165

Round 2
Reviewer 1 Report
I have reviewed the revision and made some minor suggestions.
Please see attachment.

Author Response
Dear reviewer,
Thank you again for your detailed comments in our article. We addressed all the requested modifications. Below is the detailed point by point response.
delete " and one of the most important" The sentence should read. This study was carried out in the largest hospital in Latin America, ...
We deleted this sentence
change flowcharts to flowchart
We corrected this typo
Rearrange the order of variables. Staret with age, sex, race, etiology, alcohol consumption, and then go by physiologic variables by system
We changed the order
change to "number of brain radiographic injuries"
We changed to "number of brain radiographic injuries"
Reviewer 3 Report
I have read the article several times because I find the subject very interesting and it explores one of the areas of neurointensive care that concerns a major public health problem in many countries around the world, such as TBI.
The modifications made to the manuscript correspond to what was previously requested.
Author Response
Thank you very much for your comments and revision. We corrected a few more typos and language mistakes.
Reviewer 4 Report
The authors significantly improved the quality of the manuscript and addressed my previous concerns. I no longer have any major criticisms of the manuscript.
Author Response

(The authors gave the same response as above.)
